# Optoelectronic Strain-Measurement System Demonstrated on Scaled-Down Flywheels

**DOI:** 10.3390/s24134292

**Published:** 2024-07-01

**Authors:** Matthias Franz Rath, Christof Birgel, Armin Buchroithner, Bernhard Schweighofer, Hannes Wegleiter

**Affiliations:** 1CD-Laboratory for Measurement Systems for Harsh Operating Conditions, Graz University of Technology, 8010 Graz, Austriawegleiter@tugraz.at (H.W.); 2Institute of Electrical Measurement and Sensor Systems, Graz University of Technology, 8010 Graz, Austria

**Keywords:** flywheel, contactless strain measurement, laser

## Abstract

Monitoring the strain in the rotating flywheel in a kinetic energy storage system is important for safe operation and for the investigation of long-term effects in composite materials like carbon-fiber-reinforced plastics. An optoelectronic strain-measurement system for contactless deformation and position monitoring of a flywheel was investigated. The system consists of multiple optical sensors measuring the local relative in-plane displacement of the flywheel rotor. A special reflective pattern, which is necessary to interact with the sensors, was applied to the surface of the rotor. Combining the measurements from multiple sensors makes it possible to distinguish between the deformation and in-plane displacement of the flywheel. The sensor system was evaluated using a low-speed steel rotor for single-sensor performance investigation as well as a scaled-down high-speed rotor made from PVC plastic. The PVC rotor exhibits more deformation due to centrifugal stresses than a steel or aluminum rotor of the same dimensions, which allows experimental measurements at a smaller flywheel scale as well as a lower rotation speed. Deformation measurements were compared to expected deformation from calculations. The influence of sensor distance was investigated. Deformation and position measurements as well as derived imbalance measurements were demonstrated.

## 1. Introduction

Flywheel energy storage systems (FESS) are increasingly being used for applications in grid stability and load balancing as part of the effort to shift energy production to renewable sources [1]. One of the main challenges is the mismatch in peak energy consumption periods and production periods of wind and solar energy. Energy storage technologies are an important tool to help mitigate this mismatch. A FESS is an electromechanical system consisting of a rotating mass called the flywheel, which is attached to an electrical machine operating as either motor or generator [2]. The FESS is charged by accelerating the flywheel, converting electrical energy to rotational mechanical energy. To extract energy, the electrical machine operates as a generator, decelerating the flywheel. Important features of a FESS are a high charge/discharge cycle count, no capacity decline over the lifespan, and the ability to dimension the storage system’s capacity largely independently from its power. On the downside, the achievable energy densities are low (e.g., 30 Wh/kg, depending on material), limiting the practical capacity, and mechanical losses are high, discharging the FESS in a matter of hours. These properties are complementary to chemical battery storage systems where cycle count is usually a limiting factor, but self-discharge is less of a concern. This makes FESS interesting for applications in small-scale grid stability, acting as a local, short-term energy buffer for high-power-demand applications, as found in public transport electric vehicle charging stations or industries with brief power draw peaks [3]. Since the energy capacity of a FESS is proportional to the square of the flywheel’s rotation speed, we are interested in selecting materials that withstand high centrifugal stresses. Carbon-fiber-reinforced plastic (CFRP) offers higher tensile strength compared to steel and can be used to selectively reinforce a steel flywheel by press-fitting outer rings of hoop-wound CFRP over a steel shaft incorporating the bearing seats.

The composite construction of such a flywheel offers unique manufacturing challenges and requires specialized methods of condition monitoring. For instance, to monitor the interface between multiple press-fitted CFRP hoops, strain has to be measured locally at multiple radial positions [4].

A scaled-down test was performed to verify the sampling-speed requirements of the sensor design as well as to generate test data of the whole measurement setup for further simulation verification. This paper presents preliminary experimental results of a contactless, optoelectronic strain-measurement system, applied to a scaled-down laboratory model of a flywheel system. The system measures the position and deformation of the flywheel rotor in operation and can be used to identify operation parameters like imbalance, critical resonant operation speed, and directional deformation of the flywheel.

Previous papers focused on the influences that contribute to the overall measurement uncertainty from the optical parts of the sensor system as well as influences from flywheel movement [5,6]. This work is an extension and continuation with a focus on the implementation and testing of the complete measurement system on a miniaturized test bench.

## 2. Optoelectronic Strain Measurement Sensor

This section offers an introduction to the operation principle of the optoelectronic strain measurement (OESM) system and describes the design of a single sensor unit as well as the arrangement of multiple sensors in a sensor system. A reflective paint pattern is required to interact with the optical sensors. The application process of this reflective pattern to the rotor surface is shown. The adjustment process for a single sensor as well as the electrical connection and data acquisition are described.

### 2.1. Introduction to Optoelectronic Strain Measurement

This section is cited from a paper of previous work covering the same OESM system [5]. It explains the principle operation of an optoelectronic strain measurement (OESM) method for flywheel rotors as shown in Figure 1 [7]. A pattern of spiral sections is applied to the top surface of the rotor. This pattern is locally illuminated in a small spot by a stationary light source, and the reflected light is measured by a sensor. The intensity of the reflected light depends on the surface it is reflected from. The dark CFRP surface reflects less light than the chrome paint used to apply the pattern.

Over each full rotation of the flywheel, a characteristic sequence of light and dark pulses is measured along the circumference of the pattern, as illustrated in Figure 2. Strain in the rotor, for instance from centrifugal forces, causes the rotor and also the applied pattern to deform and stretch radially. Due to this deformation, the stationary light source and sensor illuminate a slightly altered shape of the pattern, changing the relation of light pulses received by the photodiode. From the duty cycle of the light pulses, the position on the pattern and therefore the relative deformation of the flywheel can be calculated.

The special pattern, shown in Figure 1, consists of lobes: filled, three-edged shapes where two edges are pieces of Archimedean spirals [8]. In polar coordinates, an Archimedean spiral is described by its constant relation of the radius *r* and the angle θ. Lobes are repeated over the circumference and radius, providing multiple measurement positions on the flywheel.

For the example with CFRP base material as in Figure 1 the duty cycle α can be described as follows:(1)$$\alpha =\frac{t_{chrome}}{t_{lobe}}=\frac{t_{chrome}}{t_{chrome}+t_{CFRP}}$$
where $t_{chrome}$ is the measured width of the light pulse, $t_{CFRP}$ is the width of the dark pulse, and $t_{lobe}$ is the width of one lobe.

A more general form of Equation (Equation 1), for any combinations of base and pattern material, is
(2)$$\alpha =\frac{t_{pattern}}{t_{lobe}}=\frac{t_{pattern}}{t_{pattern}+t_{base}}$$
where $t_{pattern}$ is the measured width of the painted pattern, $t_{base}$ is the width of the flywheel base material, and $t_{lobe}$ is the width of one lobe.

To measure the deformation *u* at a sensor position, we have to determine its radial position within a lobe from the respective duty cycle measurement α. The radial position on a lobe segment *r* can be calculated as
(3)$$r=r_{o}-\alpha \;\cdot \;\left(r_{o}-r_{i}\right)$$
where $r_{o}$ and $r_{i}$ are the outer and inner radii of the lobe segment within the detectable range of the sensor. Reference values for each sensor position are acquired at low rotation speeds, where the deformation is negligibly small. The local deformation *u* at each sensor position can then be determined by the difference in the reference value and the instantaneous measurement.

### 2.2. Sensor Design

The OESM sensor system requires multiple optical sensor units for its operation. A single OESM sensor unit consists of a laser light source, a photodiode, and a 3D-printed frame to hold the components in place as well as allow mounting. The sensor schematic is shown in Figure 3. The main criterion by which the photodiode (OP906, TT Electronics [9]) was chosen was its short rise time of 5 ns so the fast transients of the reflected PWM-signal can be captured without additional distortion from the photodiode.

The laser light source is an off-the-shelf laser pointer module (DI605-1-3(8x21)-ADJ, Picotronic [10]) with an integrated laser diode, lens, focusing mechanism, and driver circuit and is supplied with 3 V. It is cylindrical with 8 mm diameter at 21 mm length. This module was chosen because it allows the sensor to be compact while still offering an adjustable focus to change the size of the laser spot. The laser module is friction-fit at its base end in the frame. The laser’s front end can be adjusted with four set screws for fine-tuning its position. This provides a method to aim the laser and adjust its reflection to directly hit the photodiode.

The jig, in which a single OESM sensor can be adjusted, is shown in Figure 4. The sensor was mounted at a reference distance of 6 mm from the photodiode and is initially aimed at a black surface to adjust the laser focus. The resulting laser spot sizes and corresponding focus settings were recorded with a laser beam profiler. An exemplary beam profile is shown in Figure 5. To aim the reflected laser beam at the photodiode, the sensor was mounted at reference distance above a surface-coated mirror, and the set screws were adjusted until the beam hit the photodiode’s center.

### 2.3. Measurement Setup

The electrical setup for one sensor is shown in Figure 6. Each sensor’s photodiode is connected in reverse to the supply voltage $V_{sup}=36\;V$. The resulting diode current $I_{PD}$ is proportional to the light intensity. The shunt resistor $R_{shunt}$ influences the measurable rise time of the photodiode signal. A larger resistance value will result in higher voltage signal amplitudes at the oscilloscope but will limit the signal bandwidth. Smaller values will in turn lead to a smaller signal amplitude and higher bandwidth. For the measurements in this paper, a resistance $R_{shunt}=10\;k\Omega$ is used as a good trade-off between signal-to-noise ratio and signal rise time.

The oscilloscope used to measure the photodiode signal is a PicoScope 4444 with up to 20 MHz bandwidth and up to 14-bit resolution. It connects to a PC via USB and is controlled by a custom Matlab measurement script. Due to its design and material, our flywheel is limited to a rotational speed of 18 krpm (300 Hz). Hence, the bandwidth of the PicoScope 4444 is sufficient. As presented in a previous paper, a time resolution of 3.3 ns is desirable while interpolation from lower sample rates introduces additional uncertainty in the measured flank position in the photodiode signal and therefore in the duty cycle α [5].

### 2.4. Pattern Design

A specific pattern consisting of pieces of Archimedean spirals was applied to the flywheel rotor’s surface via a process of masking and spray painting. The pattern has to create a contrast to the natural base color of the flywheel rotor’s material. Reflective properties of relevant materials were presented in a previous paper [11]. For steel flywheels that are naturally reflective, black paint was used. For the PVC flywheel, which in this specific case has a gray base color, a reflective silver paint was used. A pattern with four repeating lobes, as shown in Figure 7, allows four strain and position measurements during each flywheel revolution. The pattern was created by a Matlab script and exported as vector graphic. From the pattern file, a paint mask was cut out of vinyl foil with a vinyl cutter (Ritrama O-400 foil, Roland GX-24 vinyl cutter).

Each lobe, in addition to the spiral pieces, also has a patch of constant angular width for reference purposes. For one lobe, the calibration patch is interleaved with a notch which is used as the index reference. This allows the assignment of the PWM data from the measurement signal stream to a specific lobe and therefore a specific position and orientation on the flywheel. The repeated rings of the pattern were 10 mm wide, with the outermost 2 mm consisting of a continuous ring to hold the cut-out vinyl foil pieces together when they were applied to the rotor surface. This area cannot be used for sensing, which is an acceptable tradeoff for ease of pattern mask application. Since the laser beam was aimed at the ring’s center position (5 mm from the inner ring radius), this leaves a measurement range of −5 mm to +3 mm.

After cutting, the vinyl foil was still attached to a so-called carrier foil. Areas of the pattern that should be covered in paint had to be manually removed from the foil mask. Afterward, the foil mask was covered in a transfer foil, the carrier foil was removed, and the mask was applied to the flywheel. The surface of the flywheel was wetted (water for PVC, oil spray for steel) to aid the positioning of the mask. After drying, the transfer foil was removed, and the surface was cleaned and then spray-painted. Positioning of the pattern on the flywheel was identified as a source of inaccuracy, so care has to be taken in the manual application process. Because of the limited precision further compensation might be necessary [6].

### 2.5. Sensor System and Positioning

One single sensor can fundamentally only measure the shift of the pattern relative to the sensor position. A rotating flywheel will exhibit deformation and displacement due to rotor dynamics. Displacement can occur perpendicularly to the rotation axis (in-plane displacement, in the plane of the pattern on the circular face of the cylindrical flywheel) or along the rotation axis (out-of-plane). The deformation due to centrifugal force is a function of speed and radial position on the flywheel. An OESM sensor looking at a piece of the pattern on the flywheel will generally measure a combination of the flywheel’s in-plane displacement and deformation. By deploying multiple sensors at different positions, it is possible to separate the general behavior of the flywheel into the displacement part and the deformation part. Orthogonal sensor placement further enables the determination of the direction of in-plane displacement.

The placement of sensors is shown in Figure 8 where the letters A, B, C, and D indicate the angular position, and the subscript corresponds to the ring on the pattern the sensor is centered on. Numbering starts at 1 with the outermost ring and continues inward.

From the raw voltage signal, measured at the photodiode’s shunt resistors, pulse width values were calculated. From the measured pulse widths, the duty cycle is calculated according to Equation (Equation 2). The duty cycle then translates to a shift between the pattern and the sensor position according to Equation (Equation 3). The uncalibrated shift value is the sum of the sensor placement error, the pattern placement error, the in-plane displacement of the flywheel, and the deformation of the flywheel. Sensor and pattern placement errors can be compensated by calibrating to a reference position, recorded at low rotation speeds (i.e., the lowest speed still recorded in a coast-down test before standstill). During this calibration run, deformation due to centrifugal forces will be negligibly small, and the axis of rotation is dictated by the flywheel bearing.

All sensor readings during this calibration phase are therefore offset due to non-perfect sensor placement or non-perfect pattern placement and are subtracted from subsequent measurements. Deviations in sensor placement and in paint pattern were further investigated in a previous paper [6].

Because of the linear relation between the pattern shift relative to the sensor and the pulse width of the measured signal, a homogeneous deformation of the flywheel leads to the same increase in pattern shift measurement on all sensor positions of the same radius. For small in-plane displacements of the flywheel, we can neglect the influence of the non-linear relation between radial position and deformation. The mean pattern shift value of a pair of sensors on opposing sides of the flywheel corresponds to the deformation along the sensor pair axis. Sensor pair A-B in Figure 8 corresponds to the deformation in the y-direction while sensor pair C-D corresponds to the deformation in the x-direction.
(4)$$u_{x}\left(i\right)=\frac{s_{C}\left(i\right)+s_{D}\left(i\right)}{2}$$
(5)$$u_{y}\left(i\right)=\frac{s_{B}\left(i\right)+s_{A}\left(i\right)}{2}$$

In-plane displacement of the flywheel along an axis will lead to inverse relative change in the sensor values of the respective sensor pair (A-B for the y-axis, C-D for the x-axis). Half the difference in sensor values of a pair corresponds to the flywheel in-plane displacement in the respective axis direction.
(6)$$d_{x}\left(i\right)=\frac{s_{C}\left(i\right)-s_{D}\left(i\right)}{2}$$
(7)$$d_{y}\left(i\right)=\frac{s_{B}\left(i\right)-s_{A}\left(i\right)}{2}$$

Measured shift values *s* for each sensor are assigned a time index *i* where every time step corresponds to a shift measurement over one lobe of the pattern. For a pattern with four lobes, a full rotation of the flywheel can be resolved in four time steps. When calculating deformation and in-plane displacement, care has to be taken to use the shift values from the same time index *i*. The index mark (double line in the reflective pattern) is used as a reference to assign the correct timing index to shift measurements from different sensors.

## 3. Experimental Test Bench Setup

The OESM setup was tested on two different flywheel test benches: An open test bench was used for low-speed tests where the sensor position can be varied by a stepper motor assembly. For high-speed tests, a closed test bench was used. In case of a rotor burst because of material failure or adverse operating conditions, it can withstand the impact of a PVC flywheel.

### 3.1. Low-Speed Test Bench for Single-Sensor Characterization

The low-speed test bench consists of a 27 kg steel rotor (25 cm diameter, 7 cm height) and is designed for a maximum rotation speed of 40 Hz. A photo of the whole test bench is shown in Figure 9. Since the rotor surface is naturally reflective, the pattern was applied in black spray paint to create an optical contrast. The masking and painting process as well as the pattern itself are described in Section 2.4. The rotor with the applied pattern can be seen in Figure 9. In order to record the characteristic curve of the OESM sensor, a single sensor was mounted on a stepper motor translation stage with µm positioning resolution to adjust the sensor in the radial direction of the flywheel. The sensor was moved to its nominal distance of 6 mm from the rotor surface and aimed at the center of one ring section of the pattern.

The characteristic curve of the OESM sensor describes the relation of the radial shift between the sensor and pattern on the surface and the photodiode signal’s relative pulse width. For the measurement series of the characteristic curve, the flywheel was held at a constant speed of 40 Hz. The sensor was radially moved by the translation stage relative to the pattern, and the sensor’s photodiode signal was recorded over multiple flywheel revolutions at each position. For increased accuracy through averaging and uncertainty analysis, the whole cycle was repeated multiple times and also repeated with increasingly finer position increments.

To analyze the influence of axial distance between the OESM sensor and the flywheel, a second translation stage was added to change the sensor’s distance from the surface as well. At the nominal distance of 6 mm, the laser beam, reflected on a mirror surface, would hit the center of the photodiode. Since the laser beam behaves like a Gaussian beam, its diameter increases with distance from the laser source, and the intensity over its cross-section follows a two-dimensional Gaussian function. In its nominal position, the reflected beam is centered on the photodiode. A change in sensor distance also causes a parallel translation of the reflected beam from the the surface. The parallel translation changes the area of the Gaussian beam, which overlaps with the photosensitive area of the photodiode while the change in beam length influences the intensity.

The laser focus was set to achieve a beam spot width of ∼500 μm at a nominal sensor distance. This provides a balance between transition detection speed (smaller spot) and reduction in surface inhomogeneity noise (larger spot).

### 3.2. High-Speed Test Bench with PVC Flywheel for Sensor System Evaluation

The high-speed test bench was used to evaluate the OESM system at faster revolution speeds during which the flywheel deforms due to centrifugal force. A flywheel made out of polyvinylchlorid (PVC) was used rather than one made from steel or aluminum because PVC (Young’s modulus E=∼3 GPa) will deform more than metal (Young’s modulus for steel E=∼210 GPa) at the same speed. Additionally, the lighter PVC flywheel’s energy content will be lower, making it less likely to damage the sensors or the test bench in the event of a malfunction, e.g., a rotor burst.

The key parts of the test bench are the motor, the magnetic coupling, and the flywheel under test in a vertical arrangement [12]. The key parts and the sensor assembly are placed in the airtight safety housing, as shown in Figure 10 [12]. The water-cooled motor is rated at 2.2 kW at up to 42 krpm. Since the flywheel needs to be operated in a vacuum to reduce air friction, the flywheel chamber can be sealed and evacuated with a vacuum pump to ambient pressures of below 1 mbar. A magnetic coupling is used to transfer the torque of the motor outside the vacuum chamber to the shaft of the flywheel. A thin, rigid fiberglass membrane separates the two pressure environments and seals the flywheel chamber.

The flywheel with a diameter of 22 cm was cut from a 9 mm PVC sheet, mounted to an axle hub, and then machined on a lathe to improve roundness and balance [13]. Since our specific PVC is dark gray in color, the OESM pattern was applied in a reflective silver paint to create a contrast for the sensor, as described in Section 2.4. The PVC flywheel is supported by two ball bearings, one above and one below the flywheel.

The sensor assembly shown in Figure 10 consists of mounting rails in a cross configuration for easy repositioning of the OESM sensors at different angular positions and radial distances as described in Section 2.5. The lid of the high-speed test bench with the mounted PVC flywheel can be seen in Figure 11.

## 4. Flywheel Mechanics

This section will explain the mechanical flywheel properties the OESM system is able to measure. Formulas used to calculate the theoretical reference values for comparison with the measurements will be introduced. We will also discuss why these properties are of interest for monitoring the condition of a flywheel energy storage system.

### 4.1. Deformation Calculation

Deformation due to centrifugal forces, when the flywheel is simplified to a uniform disk with a center hole, can be calculated as the radial deformation displacement
(8)$$u_{theory}\left(r,\omega \right)=\frac{\rho \omega^{2}}{8E}\left(\left(1-\nu \right)\left(3+\nu \right)\left(r_{hole}^{2}+r_{out}^{2}\right)r+\frac{\left(1+\nu \right)\left(3+\nu \right)r_{hole}^{2}r_{out}^{2}}{r}-\left(1-\nu^{2}\right)r^{3}\right)$$
with the material parameters density ρ, Poisson’s ratio ν, and Young’s modulus *E*. The disk’s size is described by its outer radius $r_{out}$ and the radius of the hole $r_{hole}$. Variables are the rotation speed ω and the radial position *r*. The speed ω in rad/s can be converted to *f* in rev/s or *n* in rev/min by
(9)ω=2πf;n=f·60.

The material parameters ρ and ν were taken from manufacturer data sheets, and Young’s modulus *E* for the PVC material was determined experimentally by spinning a flywheel specimen until it burst [13]. The calculated burst speed was then fit to the measured one by adjusting the value of Young’s modulus *E* in the calculation. The value for E= 2.9 GPa determined in this way was subsequently used for strain and deformation calculations.

The calculated radial deformation displacement at the maximal safe rotation speed n=18 krpm (300Hz) is shown in Figure 12 which indicates that the maximum deformation displacement occurs at the radius r= 95 mm. This radius was subsequently chosen as a suitable sensor position. For larger radii, the deformation displacement decreases again because there is less material for the outwards force to act on. Figure 13 in turn shows the expected deformation displacement at that fixed-sensor position.

We were also interested in the radial differential deformation displacement due to deformation $du_{theory}\left(r,\omega \right)/dr$ which was evaluated discretely by taking the difference in the measured shift values *s* of two OESM sensor modules a certain radial distance apart.

### 4.2. Position and Imbalance

This paragraph provides only a brief insight into the highly complex subject of rotor dynamics. In principle, the main operating modes of a rotor can be divided into subcritical and supercritical. In this context, a supercritical operating mode means an operating speed that is beyond the critical speed (i.e., 1. eigenfrequency) of the rotating system and subcritical below it. The OESM system is able to detect the center position of the spinning rotor. To explain the physics behind the in-plane displacement of the center position of a rotor, a simplified system, the Laval rotor, is used. This system consists of a vertically rotating stiff disk with a mass *m* and an eccentricity *e* attached to a massless shaft. The axis of rotation is flexible in the disk plane (xy-plane) with a system stiffness *k*. Tilting due to torque imbalance as well as friction due to relative displacements and deformation are neglected. Figure 14 shows the simplified system and the technical implementation on the high-speed test bench.

The flexible mounting of the rotor can be achieved by two main design principles shown in Figure 15a—a stiff shaft with flexible bearings, and Figure 15b—a flexible shaft with stiff bearings, and while rotating at constant speed, the equilibrium deformation of the flexible shaft does not change over one revolution, in contrast to the deformation of the flexible bearings, which undergo a complete stress cycle each rotation. The second design simplifies the implementation of damping mechanisms. In practice, this is an important property of a supercritical rotor-bearing system. However, it is not necessary to take damping into account in a first instance.

Figure 15 shows the simplified force situation related to the presented model. The acting forces in the xy-plane are the centrifugal force $F_{C}=m\omega^{2}$$r_{P}$ and the elastic force $F_{e}=r_{C}k$. For the center position $r_{C}$, the force equilibrium of the steady state operation becomes
(10)$$|r_{C}\left|=\right|\frac{m\omega^{2}e}{k-m\omega^{2}}\left|=\right|\frac{e}{\frac{k}{m\omega^{2}}-1}|$$

As shown in the graph (Figure 16), the distance $r_{C}$ becomes infinitely large at the critical speed $\omega_{cr}=\sqrt{k/m}$. At this speed, the deflection of the shaft would theoretically increase infinitely. This runout is independent of the value of the eccentricity *e* and would lead to failure of the rotating system. If we look at the supercritical range of the graph, the absolute value of in-plane displacement $r_{C}$ converges to the eccentricity *e*. This behavior is called “self-centering”; at infinite operating speed the system tends to rotate around its center of gravity. For the practical implementation of supercritical operation, the runout of the in-plane displacement during the critical speed must be prevented, and the self-centering effect must occur. Both the ability to rotate faster than the critical speed and the stability at this operation state are linked to the Coriolis acceleration [2]. Referring to Figure 15, the Coriolis force can only influence the system if point P can move freely in the xy-plane. However, in the hypothetical case that point P is restrained to move only along the x-axis in radial direction r, the rotor cannot reach a supercritical operation state [2]. In this case, self-centering cannot occur, as the Coriolis acceleration would not be able to move P in the y-direction. Even assuming that an external force places the system in its equilibrium position in the supercritical mode, such a position would still be unstable. In essence, this hypothetical case corresponds to a shaft system with different stiffness in the x and y directions, resulting in the existence of two critical speeds, each one dependent on one of the values of the stiffness of the shaft. In the operating speed range between the two critical speeds, the shaft exhibits an unstable behavior, assuming that damping can be neglected. A more detailed consideration goes beyond the scope of this paper and the reader is referred to [1].

As mentioned above, load-dependent damping *D* is implemented. This limits the maximum in-plane displacement at the resonance speed and thus reduces component loads and vibrations. Figure 16 shows the progression of $r_{C}$ related to the disk eccentricity *e* as a function of the related speed f=ω/2π with the damping *D*, as follows:(11)$$|r_{C}|=\frac{e\eta^{2}}{\sqrt{{\left(1-\eta^{2}\right)}^{2}+{\left(2D\eta \right)}^{2}}}$$

In Equation (Equation 11), η is the speed-dependent phase shift $\omega /\omega_{cr}$. In the case of weak damping, the curves of the undamped and damped implementations approximately match, with deviations only occurring in the resonance range. In the damped system, the center in-plane displacement $r_{C}$ remains finite. Depending on the level of damping, the maxima are shifted slightly to higher frequency values *f* [14].

The application of these rotor dynamic principles is important for all high-speed machines. For modern flywheel systems, the advantages of supercritical operation predominate:Lower vibration levels caused by imbalance;Reduced bearing loads;Stiffness of the shaft can be reduced;Manufacturing accuracy requirements decrease.

The OESM system enables not only the resolution of the radial rotor expansion but also the in-plane displacement of the axis of rotation. This is of great importance in terms of rotor dynamics and can help to monitor and optimize fast-spinning systems.

## 5. Results and Discussion

This section will present the results of the experimental measurements on the low-speed and high-speed test benches described in Section 3. The plot in Figure 17 gives a qualitative overview of the photodiode signal. Measurements are either performed at a constant flywheel rotation speed or as a coast-down experiment. In a coast-down experiment, the flywheel is accelerated to the target speed, after which the motor is turned off. The flywheel is then allowed to slow down to a halt on its own, decelerated only by inherent frictional forces. Because of the inertia of the flywheel, the coast-down of the PVC flywheel takes up to 2 min, over the course of which repeated measurements at decreasing speeds are taken. This results in a characteristic curve of the measured value over speed. Measurements of coast-down experiments are less prone to errors due to the interference of the running motor. The interference and noise in the unfiltered photodiode signal can result in errors in the detection of the index mark double pulse. To reduce these errors, a moving average (MA) filter with a dynamic filter length was used. In a first step, the period of one full flywheel revolution is estimated from the DFT of the signal. In a second step, the MA filter length is chosen accordingly, e.g., as 1% of the period. The threshold for the flank detection is dynamically adjusted as well by a similar approach. An MA filter with a length corresponding to two flywheel revolutions is applied to the photodiode signal to generate a threshold value, which can adjust for changes in reflected light intensity. Reflected light intensity is dependent on the homogeneity of the reflective part of the pattern. As can be seen in Figure 17, the measured reflected light intensity differs between the four pattern lobes, each represented by a corresponding pulse in the plot. The reflected light intensity can change between measurements when a shift in the pattern occurs, for instance by radial deformation of the flywheel. A dynamically set threshold can account for those changes.

### 5.1. Low-Speed Characteristic Curve

The first measurements were taken at the low-speed test bench to evaluate the basic function of the OESM system with one sensor module. The test bench setup is described in Section 3.1, and the rotation speed was kept constant at 40 Hz. Results from a sweep over ±1 mm, plotted in Figure 18, show the relation between relative sensor position and duty cycle α. The starting position 0 mm was chosen in the radial center of a lobe, and while the measured characteristic curve follows the theoretical straight line on the macroscopical scale of ±1 mm, there is a periodic ripple in the measured curve. The periodic change in the high-level mean photodiode signal value (plotted in Figure 19) reveals a systematic influence in the low-speed setup. The period of the ripple is 200 μm, which matches the width of the tool groove marks on the steel flywheel rotor surface left by the lathe machining operation. These tool grooves create a fringed edge in the paint pattern, emphasizing the importance of the edge quality for the OESM system.

### 5.2. High-Speed PVC Flywheel Test Bench

In the following sections, the results from measurements on the high-speed PVC flywheel test bench, whose components and setup are described in Section 3.1, are presented.

#### 5.2.1. Deformation

The main value of interest examined with the help of this test bench is the deformation of a PVC flywheel due to centrifugal forces. The result most accurately fitting the theoretical behavior is achieved by calculating the mean average $u_{x}\left(i\right)$ and $u_{y}\left(i\right)$ as well as averaging over all lobe position indexes *i*, as shown in Figure 20. Measurements of small deformations are relatively influenced more than larger deformations by inaccuracies of pattern position or radial in-plane displacement of the rotational axis. Hence, the curve offset is calibrated to the largest measured deformation at 293 Hz. The large deviation from the theoretical curve at 184 Hz coincides with a resonant frequency in the test bench setup, which was also audible as an increase in sound level during testing. This resonance is measured by acceleration sensors on the outside of the test bench housing.

Plotting the results for each flywheel rotation position n=[1,2,3,4] is achieved by taking the average over all measurements where the index kmod4+1=n where 4 is the number of lobes in the pattern and therefore the number of distinguishable rotational positions. Further averaging over $\bar{u_{x}}\left(n\right)$ and $\bar{u_{y}}\left(n\right)$ results in the average deformation for a given rotation position *n* plotted in Figure 21. This can be interpreted as differences in deformation as seen from a coordinate system relative to the flywheel.

Similarly, averaging over all *i* and plotting $\bar{u_{x}}\left(i\right)$ and $\bar{u_{y}}\left(i\right)$ separately results in Figure 22, showing differences in deformation relative to the cardinal sensor directions *x* and *y*. This can be interpreted as differences in deformation as seen from a coordinate system relative to the test bench and sensor assembly.

#### 5.2.2. In-Plane Displacement

The mean in-plane displacement of the pattern lobes over frequency, plotted in Figure 23, shows the movement in an arch between the calibrated zero position for low rotation speeds and the maximum speed. Lobe in-plane displacements are normalized before averaging by rotating them to the same side (Sensor A, lobe 1). This plot also includes the mean over all rotation positions which corresponds to the in-plane displacement of the rotational axis.

Plotting the absolute distance between the zero position and flywheel center position over frequency, shown in Figure 24, reveals that the flywheel transitions between two stable positions. We can further calculate the average distance of the flywheel’s imbalanced rotation around its center position which is at a maximum at 173 Hz, corresponding to the vibration maximum measured on the test bench housing.

## 6. Conclusions and Outlook

This paper has demonstrated the viability of a simple OESM system for flywheel deformation and in-plane displacement measurement. For common materials like steel and CFRP, large deformation values are only achievable with larger-radius flywheels or at higher rotation speeds. Those deformation values have been emulated by the use of a PVC flywheel in a scaled-down test bench. The deformation measurements of the PVC flywheel are in good agreement with the theoretically expected behavior. Via the position measurement, the in-plane displacement of the flywheel rotational axis over an acceleration/deceleration cycle can be tracked and the imbalance amplitude calculated. Surface quality of the flywheel as well as surface- and paint-edge quality of the reflective pattern have been identified as key contributors to measurement accuracy.

The next steps in this research are measurements on a scaled-down CFRP rotor which can be spun at a higher speed of 500 Hz, investigation of differential strain measurement capabilities, and implementation of cross-sensitivity correction algorithms.

## Figures and Tables

**Figure 1 sensors-24-04292-f001:**
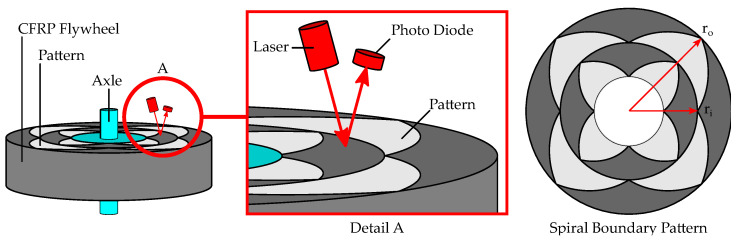
Basic principle of optoelectronic strain measurement (OESM) on a flywheel [5]. A special pattern consisting of sections of Archimedean spirals is applied to the rotor. A laser illuminates a spot on the rotor surface, and a photodiode is used to measure the reflected light.

**Figure 2 sensors-24-04292-f002:**
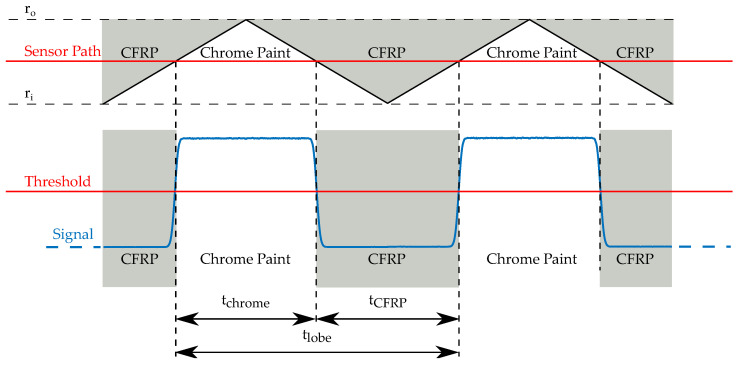
Electrical output signal from the OESM system [5]. The transition from CFRP to chrome paint is called the edge transition function and is not an ideal step function. A threshold is applied to determine the time of the transition.

**Figure 3 sensors-24-04292-f003:**
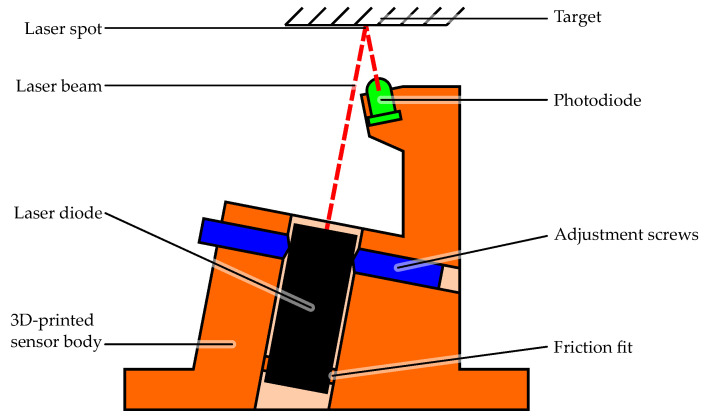
Design schematic of a single OESM sensor element.

**Figure 4 sensors-24-04292-f004:**
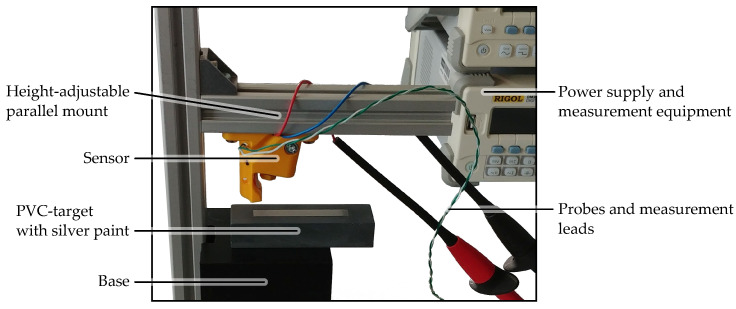
Bench-top measurement setup for the laser focus, distance, and position adjustment of a single OESM sensor. For position adjustment, the target is replaced by a mirror.

**Figure 5 sensors-24-04292-f005:**
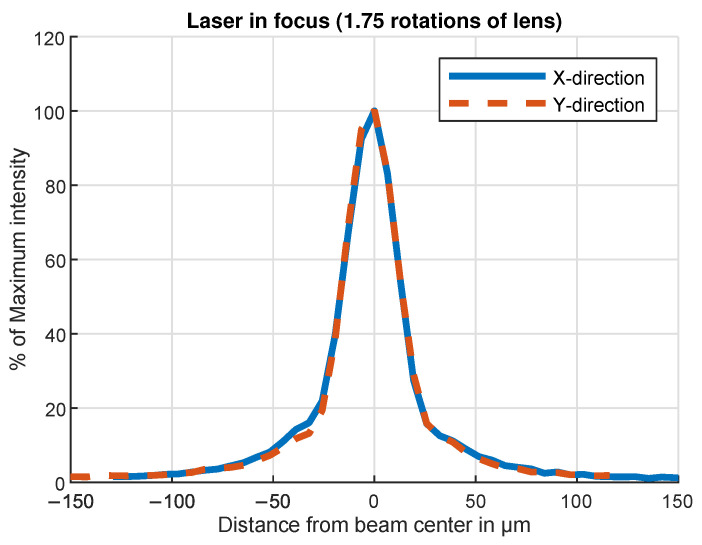
Measured beam profile of the laser light source. The focusing lens screw is set for the minimum achievable beam width.

**Figure 6 sensors-24-04292-f006:**
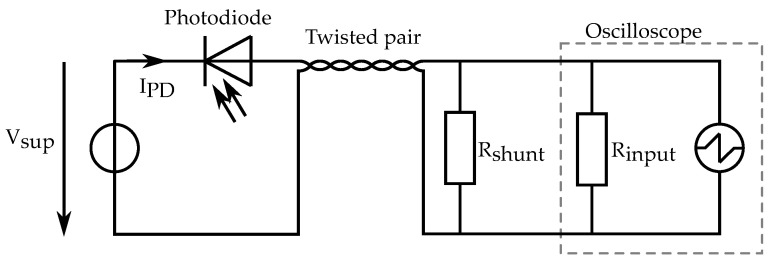
Schematic of the electrical setup of one OESM sensor.

**Figure 7 sensors-24-04292-f007:**
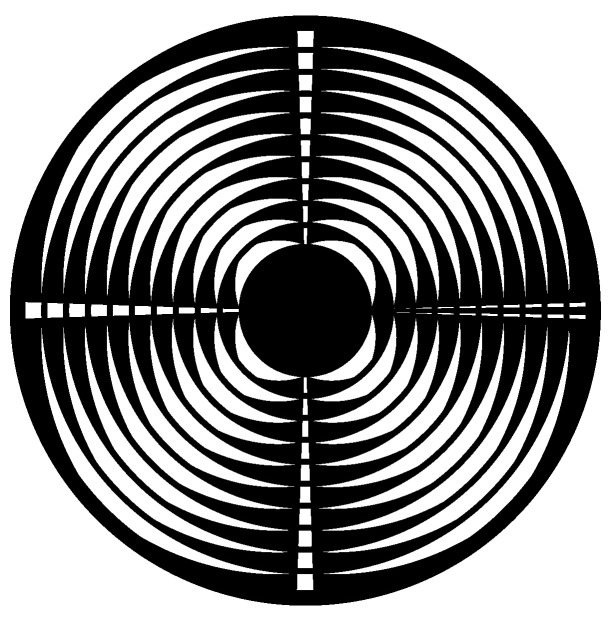
OESM pattern design. The white areas represent the reflective surface (bare metal in the case of a steel flywheel or reflective silver paint on a PVC flywheel).

**Figure 8 sensors-24-04292-f008:**
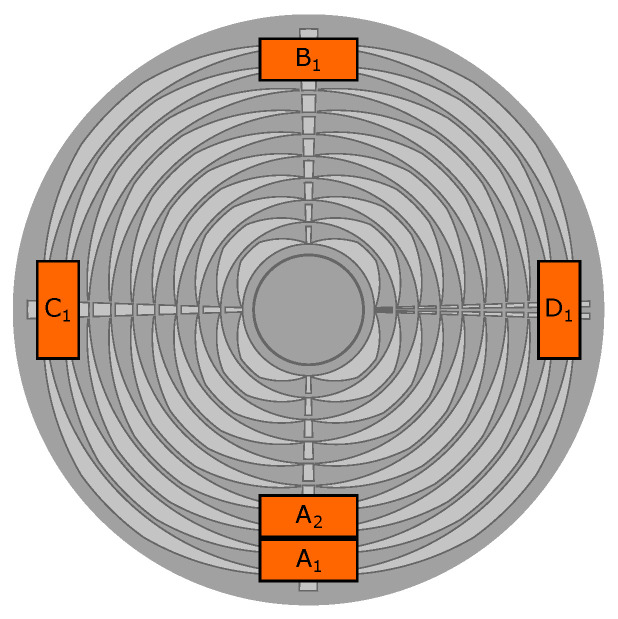
Sensor positioning on the test bench relative to the OESM pattern on the flywheel (top-down view through the flywheel).

**Figure 9 sensors-24-04292-f009:**
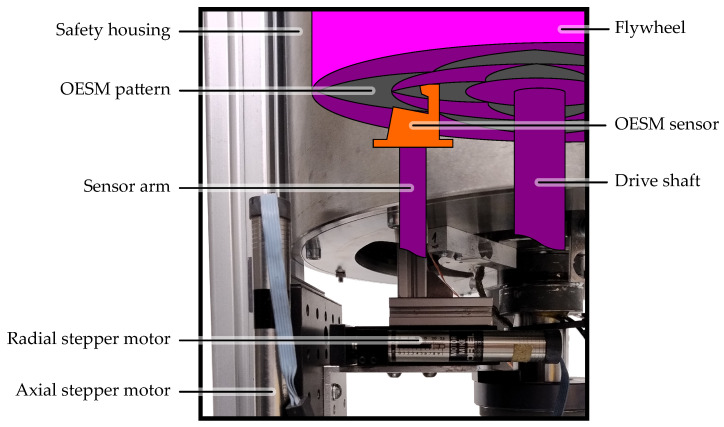
Low-speed test bench with steel flywheel. The motor (not pictured) is mounted above the flywheel and coupled via a magnetic coupling.

**Figure 10 sensors-24-04292-f010:**
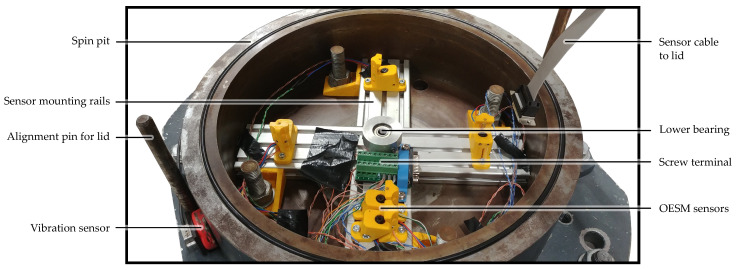
Lower half of the high-speed test bench with four OESM sensors mounted on rails. Sensor cables are attached at a screw terminal. A ribbon cable is used to connect the screw terminal to a vacuum lead-through in the test-bench lid. When the lid is closed, the flywheel drive shaft is inserted in the lower bearing for stability. The vibration sensor is used to determine the resonant frequency of the flywheel system.

**Figure 11 sensors-24-04292-f011:**
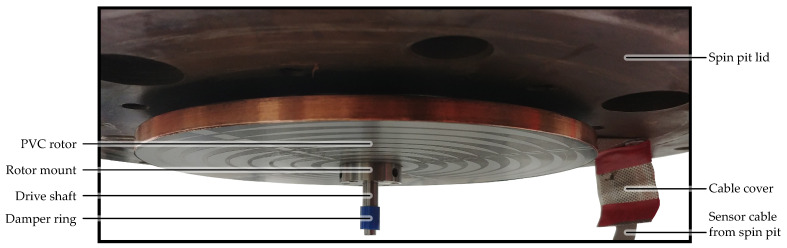
Test bench lid with mounted PVC flywheel. The motor is mounted above the flywheel and connected to the drive shaft via a magnetic coupler. This allows the flywheel to spin in a vacuum even though the motor is mounted outside the vacuum for cooling reasons.

**Figure 12 sensors-24-04292-f012:**
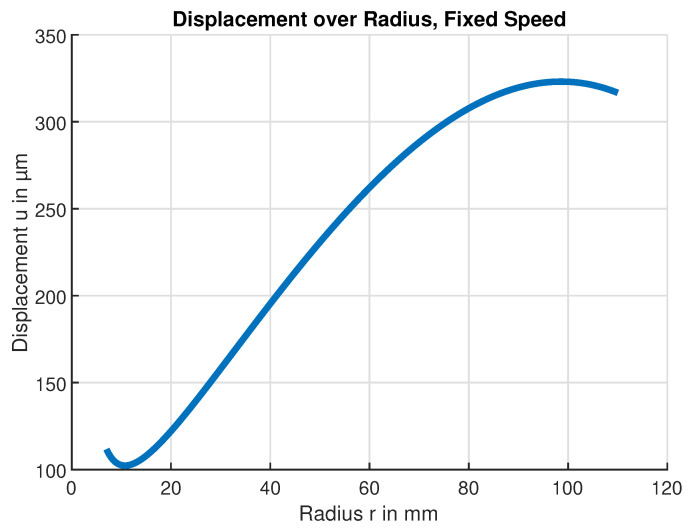
Calculated PVC flywheel deformation. Deformation displacement *u* over radius *r* for a rotational speed of f=300Hz.

**Figure 13 sensors-24-04292-f013:**
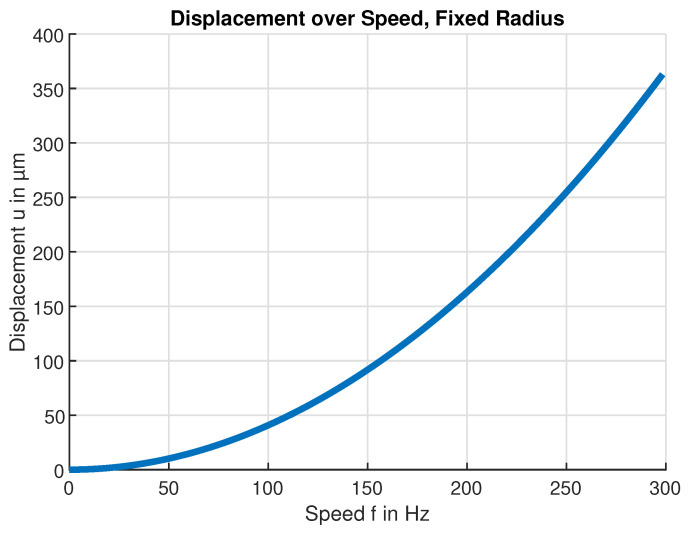
Calculated PVC flywheel deformation. Deformation displacement *u* at radius r=95mm over rotational speed *f*.

**Figure 14 sensors-24-04292-f014:**
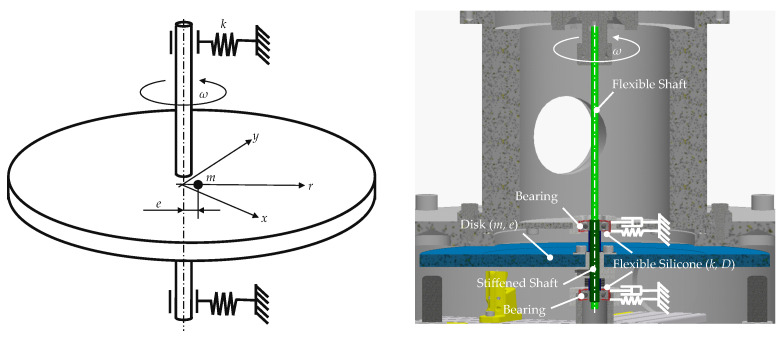
Schematic sketch of a Laval rotor (**left**), test setup (**right**).

**Figure 15 sensors-24-04292-f015:**
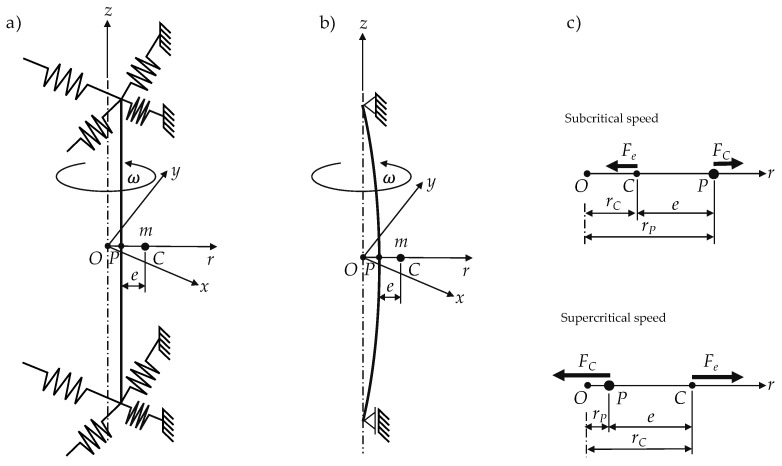
Mounting design of (**a**) a stiff shaft and flexible bearings and (**b**) stiff bearings and a flexible shaft; (**c**) acting forces at sub- and supercritical operation speed.

**Figure 16 sensors-24-04292-f016:**
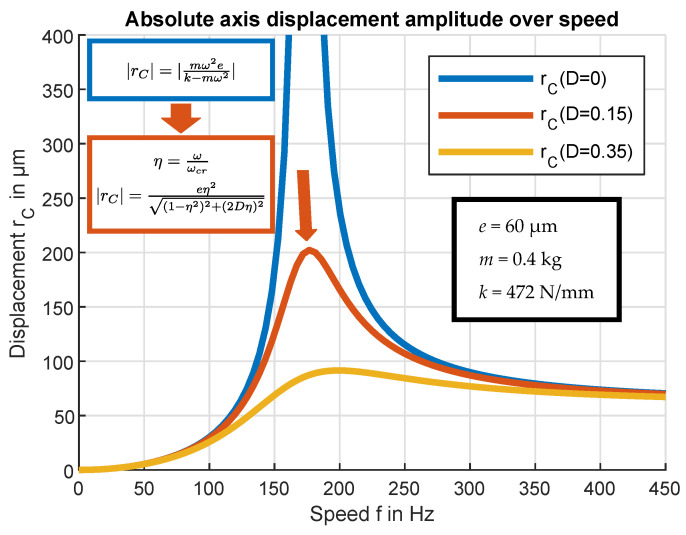
Representation of the calculated radial in-plane displacement of the rotational axis of a simplified rotor in sub- and supercritical operation mode. (top curve = undamped; middle and lower curve = damped).

**Figure 17 sensors-24-04292-f017:**
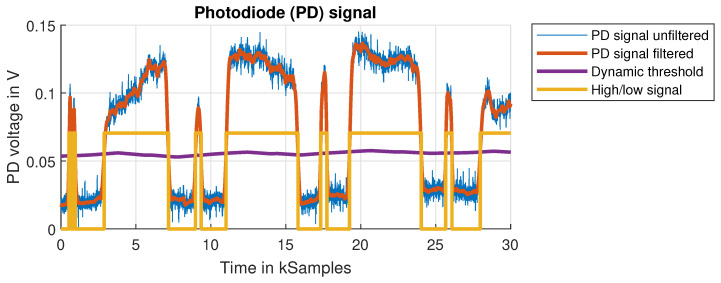
Example of the photodiode (PD) signal from an OESM sensor. Flywheel rotation speed 300Hz, sensor position A at r=95mm. The unfiltered signal shows fluctuations from the inhomogeneous reflectivity of the silver paint. A dynamic threshold is calculated to account for these fluctuations. The high and low times are determined after the threshold is applied to the PD signal.

**Figure 18 sensors-24-04292-f018:**
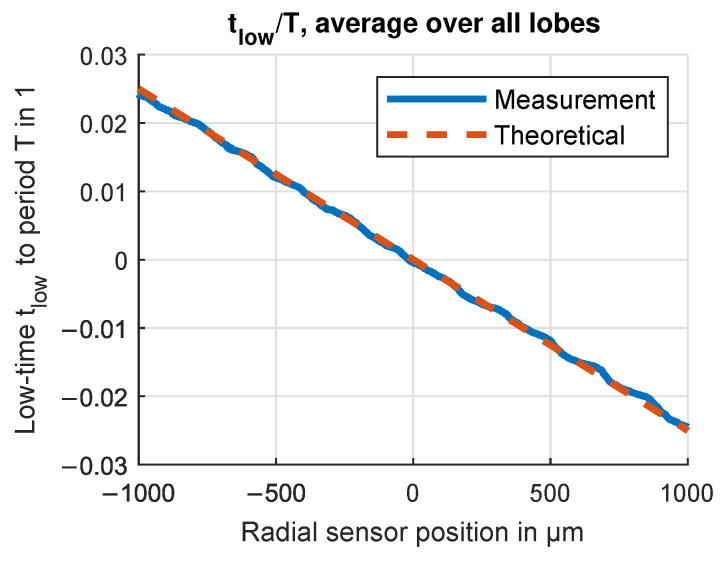
Characteristic behavior of $t_{low}/T$ over radial sensor position measured on the low-speed test bench with a steel flywheel. Values are the average over all circumferential pattern lobes. Flywheel rotation speed 40Hz, radial position r=95mm.

**Figure 19 sensors-24-04292-f019:**
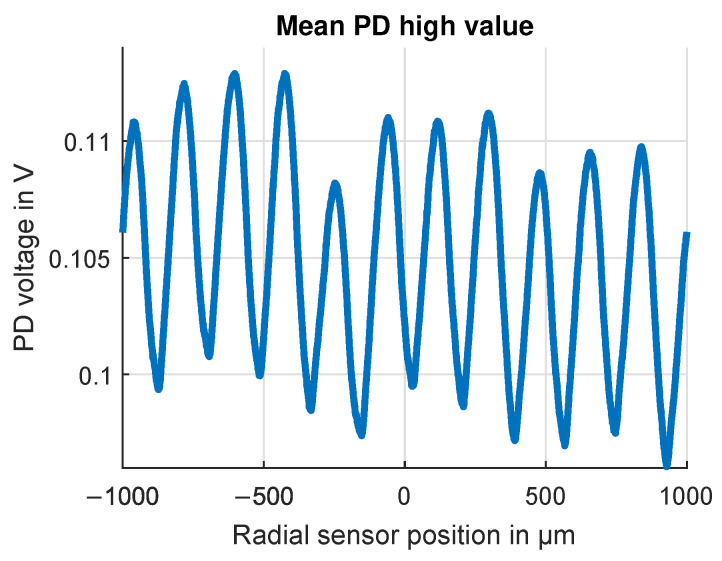
Average PD high-value voltage over radial sensor position. Only the PD voltages which correspond to areas of high reflectivity on the pattern are considered. The periodic ripple is due to periodic surface structures in the steel flywheel left by the manufacturing process. Flywheel rotation speed 40Hz, radial position r=95mm.

**Figure 20 sensors-24-04292-f020:**
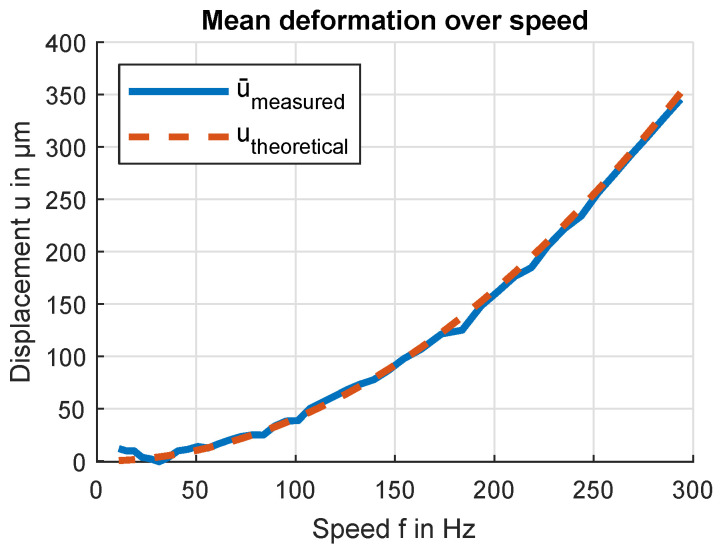
Average deformation over speed. The average value is calculated over all lobes and sensor positions. The measured curve is offset-corrected by the mean of the 10 lowest speed measurements.

**Figure 21 sensors-24-04292-f021:**
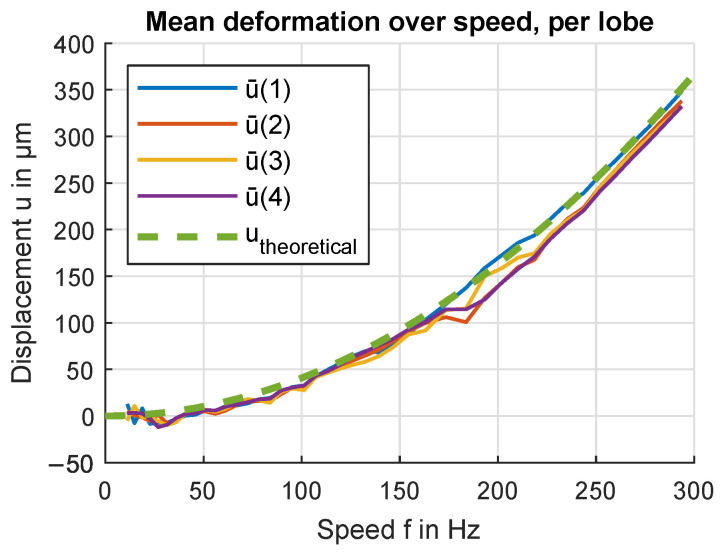
Average deformation over speed for each pattern lobe. The average value is calculated over all sensor positions. The measured curve is offset-corrected by the mean of the 10 lowest speed measurements.

**Figure 22 sensors-24-04292-f022:**
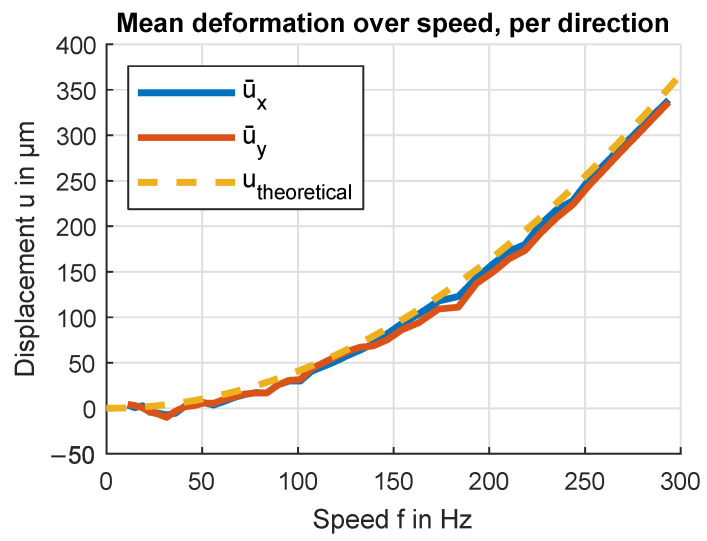
Average deformation over speed for the directions x and y. The average value is calculated over all lobes. The measured curve is offset-corrected by the mean of the 10 lowest speed measurements.

**Figure 23 sensors-24-04292-f023:**
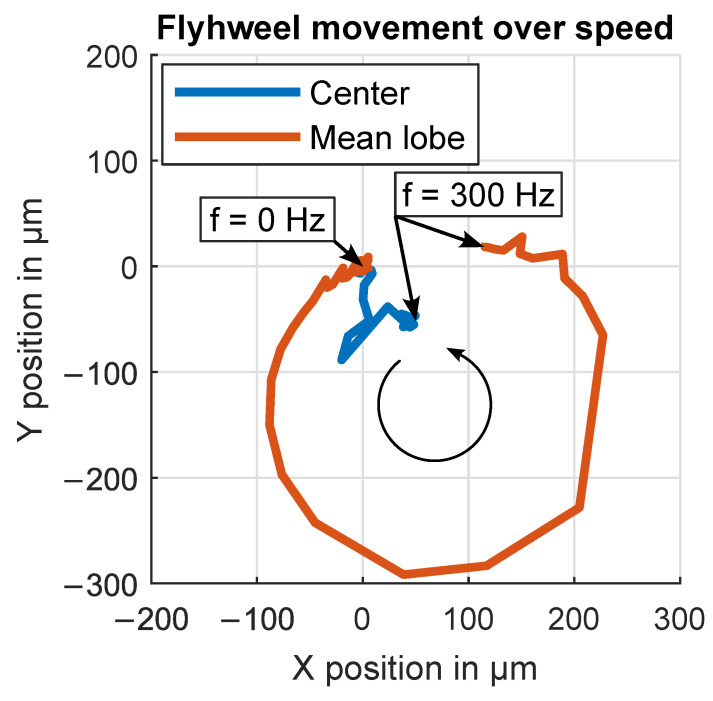
Radial in-plane displacement of rotational axis over speed in a 2D xy plot. The radial in-plane displacement of the rotational axis is calculated by averaging all the lobe in-plane displacements of a full rotation. The mean lobe in-plane displacement is calculated by rotating all the individual lobe in-plane displacements to one side (Sensor A, lobe 1) and then taking the mean.

**Figure 24 sensors-24-04292-f024:**
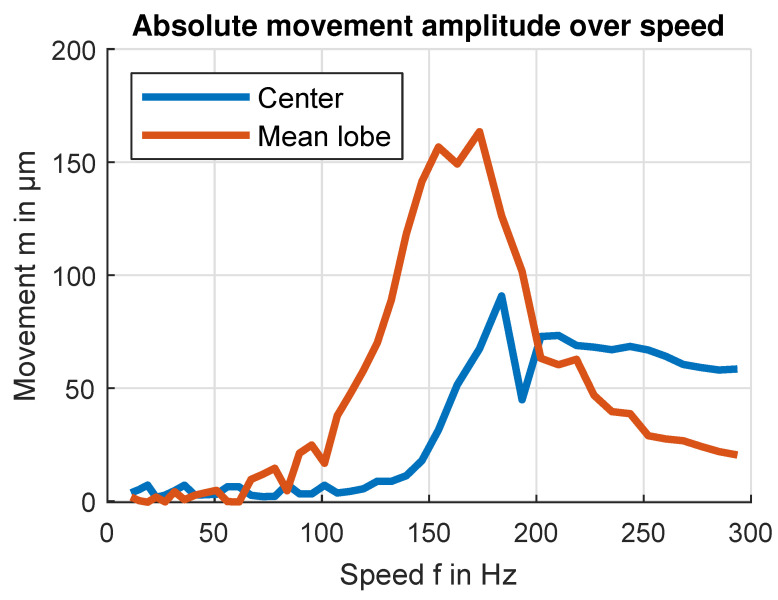
Absolute amplitude of imbalance in-plane displacement over speed. The center in-plane displacement is calculated by averaging all the lobe in-plane displacements of a full rotation. The mean lobe in-plane displacement is calculated by averaging the absolute lobe in-plane displacement amplitudes.

## Data Availability

The raw data supporting the conclusions of this article will be made available by the authors on request (matthias.rath@tugraz.at).

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
