# Peer review of "Optoelectronic Strain-Measurement System Demonstrated on Scaled-Down Flywheels"

_sensors, 2024, doi:10.3390/s24134292_

Round 1

Reviewer 1 Report

Comments and Suggestions for Authors

Author Response

Comment 1: How do sensor and pattern placement errors affect measurement results? How are these errors calibrated and compensated for?

Response 1:

Radial sensor placement errors are compensated for by calibrating them at low flywheel speeds where little to no deformation occurs. Since sensors work in pairs (AB, CD), they are calibrated as a pair. For instance, the mean of sensor values A and B at low speeds is determined. Because we assume that at low speeds no deformation occurs, the mean value must come from differences in sensor placement. This value is then subtracted from all deformation measurements in direction AB. The same procedure is applied for displacement measurements. Errors in pattern placement are the topic of an ongoing investigation and, for the time being, are corrected similarly to sensor placement errors by subtracting the offset measured at low speeds.

Added:

(196) All sensor readings during this calibration phase are therefore offsets due to non-perfect sensor placement or non-perfect pattern placement and are subtracted from subsequent measurements. Deviations in sensor placement and in paint pattern were further investigated in a previous paper.

Comment 2: In section 2.5 of the manuscript, according to what value is calculated to determine whether the higher reflection is the pattern or the substrate material? How do these values translate into offsets between the pattern and the sensor position?

Response 2:

We know from preliminary measurements that the reflective paint used to apply the pattern has more directional reflectivity than the substrate material (CFRP or PVC), effectively acting like a mirror so that the intensity of the reflected light entering the photo diode is higher than for the substrate material. The substrate has less directional reflectivity but acts more as a diffuse reflector, spreading the reflected light in all directions so that less light enters the photo diode.

The determination which photo diode shunt voltage (effectively we measure diode current) corresponds to the substrate material and which corresponds to the pattern is done with a simple dynamic threshold (Figure 17). We just have to assume that the photo diode signal will always be larger for the pattern, but that is not always the case. For instance, on a steel flywheel where the substrate (steel) has a high directional reflectivity, the pattern has to be applied in a black, non-reflective paint to achieve a contrast. Then one would have to invert the duty cycle in Equation 1.

The relation between the measured pulse widths corresponding to substrate (low) and reflective paint (high) translate into the offset between pattern and sensor position by combining Equations 1 and 2.

Added:

(84) For the example with CFRP base material as in Figure 1…

Added new Equation 2 for a more general case of material compositions.

Added:

(88) A more general form of Equation 1 for any combinations of base and pattern material, is:

\alpha = \frac{t_{pattern}}{t_{lobe}} = \frac{t_{pattern}}{t_{pattern}+t_{base}}

where t_pattern is the measured width of the painted pattern, t_base is the width of the flywheel base material and t_lobe is the width of one lobe.

Replaced:

Depending on whether the pattern or the base material is more reflective, we either calculated the low-time-to-period value or the high-time-to-period value, which translate to the shift between the pattern and the sensor position.

By:

(184) From the measured pulse widths, the duty cycle is calculated according to Equation 2. The duty cycle then translates to a shift between the pattern and the sensor position according to Equation 3.

Comment 3: According to Figure 15, it can be seen that the Coriolis force can only affect the system when the point P can move freely on the xy plane. The system that only moves radially along a line r cannot reach the supercritical operating state. What is the cause of this?

Response 3:

We have added a more detailed explanation. However, since the scope of the paper is to focus on the measurement technology, we try to make a compromise between going into the depths of rotor dynamic and describing only the most relevant effects. The hypothetical scenario described above (meaning, restricting the effect of the Coriolis force), can be studied in the listed reference.

Replaced:

Referring to Figure 15, the Coriolis forces can only influence the system if the point P can move freely in the xy-plane. Systems in which the point P only moves radially along a line r cannot reach a supercritical operation state [2].

By:

(342) Referring to Figure 15, the Coriolis force can only influence the system if point P can move freely in the xy-plane. However, in the hypothetical case that point P is restrained to move only along the x-axis in radial direction r, the rotor cannot reach a supercritical operation state [2]. In this case self-centering cannot occur, as the Coriolis acceleration would not be able to move P in the y-direction. Even assuming that an external force places the system in its equilibrium position in the supercritical mode, such a position would still be unstable. In essence, this hypothetical case corresponds to a shaft system with different stiffness in x and y direction, resulting in the existence of two critical speeds, each one dependent on one of the values of the stiffness of the shaft. In the operating speed range between the two critical speeds, the shaft exhibits an unstable behavior, assuming that damping can be neglected. A more detailed consideration goes beyond the scope of this paper and the reader is referred to [Gasch, Nordmann, Pfützner].

Reviewer 2 Report

Comments and Suggestions for Authors

The article by M. Rath et al. on an optoelectronic strain measurement system for contactless monitoring of deformation and position of scaled-down flywheels is well-organized in terms of design, setup, and analysis. It effectively addresses some strain measurement issues related to flywheels. The manuscript is suitable for publication in Sensors after minor revision.

1. In the introduction, the author discusses many issues related to energy, but it is seldom mentioned in the subsequent sections. The author can refer to the following papers for the writing style:

a.       Griffin, T., Kram, R. Penguin waddling is not wasteful. Nature 408, 929 (2000). DOI: 10.1038/35050167

b.      L.-L. Ma, C. Liu, S.-B. Wu, et al. Programmable self-propelling actuators enabled by a dynamic helical medium. Sci. Adv. 7, eabh3505 (2021). DOI: 10.1126/sciadv.abh3505

2. In lines 51-52, Page 2, the authors mentioned that ‘This work is an extension and continuation of previous work in which measurement influences were investigated on a sensor and system level’. It would be better to describe the differences between these two works.

3. In the schematic diagram of laser incidence and reflection in Figure 1, the perpendicularly incident laser illuminates the surface of the flywheel. Will it reflect at a large angle into the Photodiode? If not, please modify the schematic diagram to comply with the actual laws of optical refraction and reflection.

4. On line 86, Page 3, it would be better to include a schematic diagram in Figure 1 or other figures to describe the parameters of ro and ri to provide a clearer picture for readers.

5. As described in lines 231-233 on Page 8, does the area of the Gaussian beam change significantly with the translation distance? If so, what is the quantitative analysis of the beam spot?

Author Response

Comment 1: In the introduction, the author discusses many issues related to energy, but it is seldom mentioned in the subsequent sections. The author can refer to the following papers for the writing style:

  1. Griffin, T., Kram, R. Penguin waddling is not wasteful. Nature 408, 929 (2000). DOI: 10.1038/35050167
  2. L.-L. Ma, C. Liu, S.-B. Wu, et al. Programmable self-propelling actuators enabled by a dynamic helical medium. Sci. Adv. 7, eabh3505 (2021). DOI: 10.1126/sciadv.abh3505

Response 1:

Thank you for your suggestion, I will do so for my next publication. 

Comment 2: In lines 51-52, Page 2, the authors mentioned that ‘This work is an extension and continuation of previous work in which measurement influences were investigated on a sensor and system level’. It would be better to describe the differences between these two works.

Response 2:

Replaced:

This work is an extension and continuation of previous work in which measurement influences were investigated on a sensor and system level [5,6].

By:

(51) Previous papers focused on the influences which contribute to the overall measurement uncertainty from the optical parts of the sensor system as well as influences from flywheel movement [5,6]. This work is an extension and continuation with the focus on implementation and testing of the complete measurement system on a miniaturized test bench.

Comment 3: In the schematic diagram of laser incidence and reflection in Figure 1, the perpendicularly incident laser illuminates the surface of the flywheel. Will it reflect at a large angle into the Photodiode? If not, please modify the schematic diagram to comply with the actual laws of optical refraction and reflection.

Response 3:

The light shining on the used reflective paint is scattered in a wide area, so a distinction in reflected light levels between a black base surface and reflective paint is also possible at 0° incident angle. This was investigated in my paper [1]. However, the light intensity is higher when laser and photo diode are arranged at an angle. In the current paper, the laser and photo diode are arranged at angles of 5° from the surface normal.

The graphic in Figure 1 was intended to convey the very basic operation principle. The Figure has been changed to better reflect the actual sensor system (at 15° angles for better visibility).

Comment 4: On line 86, Page 3, it would be better to include a schematic diagram in Figure 1 or other figures to describe the parameters of ro and ri to provide a clearer picture for readers.

Response 4:

For clarification, r_o and r_i have been added to Figure 1 as well as Figure 2 which has been expanded to also include the pattern from which the signal is measured.

Comment 5: As described in lines 231-233 on Page 8, does the area of the Gaussian beam change significantly with the translation distance? If so, what is the quantitative analysis of the beam spot?

Response 5:

The beam width/profile has been measured at nominal distance only, but for different focus lens settings. The width of the spot influences the “specificity” of the sensor – while a smaller spot will lead to a crisper transition between reflective surfaces (e.g. CFRP to reflective paint), the sensor reading will also be more noisy due to inhomogeneities in the reflective surfaces. A larger spot will exert an averaging effect – the noise due to inhomogeneities is reduced, but the transition will also get blurred. The spot size was set to ~500 µm.

Added:

(254) The laser focus was set to achieve a beam spot width of ~500 µm at nominal sensor distance. This provides a balance between transition detection speed (smaller spot) and reduction of surface inhomogeneity noise (larger spot).
